# Resolving the Unusual Gate Leakage Currents of Thin-Film Transistors with Single-Walled Carbon-Nanotube-Based Active Layers

Sean F. Romanuik [1], Bishakh Rout [2], Pierre-Luc Girard-Lauriault [2] and Sharmistha Bhadra [1,*]

1   Electrical and Computer Engineering, McGill University, Montreal, QC H3A 0E9, Canada
2   Chemical Engineering, McGill University, Montreal, QC H3A 0C5, Canada
*   Correspondence: sharmistha.bhadra@mcgill.ca

**Abstract:** Solution-processed single-walled carbon nanotube (SWCNT) thin-film transistors (TFTs) in the research stage often have large active areas. This results in unusual gate leakage currents with high magnitudes that vary with applied voltages. In this paper, we report an improved structure for solution-processed SWCNT-based TFTs. The unusual gate leakage current in the improved structure is resolved by patterning the SWCNT active layer to confine it to the channel region. For comparative purposes, this improved structure is compared to a traditional structure whose unpatterned SWCNT active layer expands well beyond the channel region. As TFT performance also varies with oxide layer thickness, 90 nm and 300 nm thick oxides were considered. The improved TFTs have gate leakage currents far lower than the traditional TFT with the same dimensions (aside from the unpatterned active area). Moreover, the unusual variation in gate leakage current with applied voltages is resolved. Patterning the SWCNT layer, increasing the oxide thickness, and reducing the top electrode length all help prevent a rapid dielectric breakdown. To take advantage of solution-based fabrication processes, the active layer and electrodes of our TFTs were fabricated with solution-based depositions. The performance of the TFT can be further improved in the future by increasing SWCNT solution incubation time and reducing channel size.

**Keywords:** gate leakage current; printed electrodes; printed electronics; single-walled carbon nanotubes; solution processes; thin-film transistor

## 1. Introduction

Thin-film transistors (TFTs) are field effect transistors (FETs) that are fabricated by depositing thin films of active semiconducting layers, conductive electrodes, and dielectric layers atop an insulating substrate [1]. The most frequent and important application for TFTs is as switches for display products [2–5]. TFTs typically use either amorphous silicon or polysilicon as their semiconducting active layer, both of which require fabrication processes involving high temperatures [6]. Because of these required high temperature fabrication processes, amorphous silicon and polysilicon are both incompatible with alternative low-temperature solution-based fabrication processes [6]. Even with a continuous-wave green laser used to anneal a polysilicon film, to realize the targeted number of interface states and bulk trap sites within the polysilicon channel [7], fabrication of a TFT using such a polysilicon active layer still involves high temperature processes. Simple, low-cost, and high-throughput solution-based fabrication processes, including printing technology, possess great potential for creating TFTs using alternative active layer materials. These processes are also compatible with flexible substrates, such as Kapton polyimide, as they do not require high temperature processing. This compatibility affords a wide array of novel applications for TFTs, such as being critical components in chemical gas sensors [8,9] and biomedical sensors [10,11]. Amorphous silicon and polysilicon are incompatible with such low-temperature solution-based fabrication processes. Although organic semiconductors

can be deposited with low-temperature solution-based processes as an alternative to the traditional inorganic TFTs with amorphous silicon or polysilicon active layers, such alternative organic TFTs are inferior as they possess low carrier mobilities and are unstable in air [12,13]. An alternative active layer material that does not require high temperature fabrication processes, possesses high carrier mobilities, and is stable in air is highly desirable.

Semiconducting single-walled carbon nanotubes (SWCNTs) are a proven active layer material affording high mobilities, high on/off ratios, small operating voltages, and chemical stability [14–17]. A dense SWCNT network spanning the TFT's electrodes, with many SWCNT-to-SWCNT junctions within that network, is essential to forming a suitable active layer for a TFT. The extreme of a SWCNT so sparse as to not form an interconnected path of SWCNTs spanning the device's electrodes will not yield a functional TFT. Conversely, a dense SWCNT network with many SWCNT-to-SWCNT junctions will yield a functioning TFT with a high on/off ratio and a high mobility [14]. In addition to increasing the density of the SWCNT network, increasing the average length of the individual SWCNTs comprising that network will also result in a greater number of SWCNT-to-SWCNT junctions and as such a greater charge carrier mobility [14]. Hypothetically, the mobility of a uniform network of SWCNTs could be as large as $10^4$ cm$^2$ V$^{-1}$ s$^{-1}$. However, more practically, the reported mobilities of SWCNT-based flexible devices are typically in the 10–100 cm$^2$ V$^{-1}$ s$^{-1}$ range, as a limitation arising from the random distribution of the deposited SWCNTs [18]. SWCNT TFTs with such mobilities typically have on/off ratios in the 10–$10^6$ range [19]. SWCNT-based active layers can be deposited via a dry process such as Chemical Vapor Deposition (CVD) [20], which results in semiconducting SWCNTs with a good alignment between source and drain electrodes [21]. Dense SWCNT networks with a good alignment deposited via CVD can yield high performance devices [21]. However, CVD involves high temperatures and as such are incompatible with flexible substrates such as Kapton polyimide [22]. As a low-temperature alternative, SWCNT-based active layers can also be created via simple and cost-effective solution-based depositions that are ideal for printed electronics, including printing [23] and such solution-based coating techniques [24] as spray coating [25] and drop-casting [26]. Although SWCNT TFTs have been used for various applications, including sensors and logic gates, most contain conventionally microfabricated electrodes that negate the advantages of low-temperature and low-cost solution-based fabrication processes [15,27,28]. Moreover, devices based upon SWCNT active layers exhibit poor reproducibility, due to difficulties in controlling the formation of SWCNT networks over large areas as well as limitations concerning the synthesis of homogeneous structures [29].

For greater ease of manufacturing, solution-processed SWCNT TFTs in the research stage are traditionally fabricated without patterning the active layer [30,31], resulting in rather large active areas as compared to the source (S) and drain (D) electrodes. Having an active area larger than the S and D electrodes can negatively affect TFT performance as it introduces the complications of fringe electric fields and parasitic capacitances [32–35]. Such large active areas can also result in unusual gate leakage currents that both possess large magnitudes and vary with applied bias voltages, deviating from the expectations for a TFT in which the gate leakage current is both small and does not vary with applied bias voltages [32,36–38]. In fact, theoretically the gate leakage currents should be completely blocked, regardless of the size of the active area, in conventional bottom-gate TFT structures, as the thick dielectric layer electrically isolates the gate (G) electrode from the S and D electrodes [32]. However, unusual gate leakage currents defying these theoretical TFT expectations are often observed when the active layer is exceptionally large. This unusual gate leakage current behavior is now known to be the result of the conduction of electrons through defects and trap sites inside the dielectric layer, with larger active layer areas encompassing a greater volume of such defects and trap sites [32]. A high density of trap sites at the interface between an organic active layer (e.g., SWCNTs) and an inorganic dielectric (e.g., SiO$_2$) is moreover known to result in hysteresis in the response of TFTs fabricated with such an interface [8].

The thickness of the oxide layer ($t_{ox}$) within a TFT is also extremely important to its functionality and performance. Thinner dielectric layers also produce gate leakage currents with larger magnitudes [32]. Moreover, thin dielectrics are also prone to rapid dielectric breakdowns, due to the strong induced electric fields within the dielectric as well as the build-up of defects and ejected electrode ions during percolation creating conductive paths bridging the dielectric [39,40]. Conversely, thicker dielectrics result in higher threshold voltages. Although TFTs with $SiO_2$ as thin as 50 nm have been reported [14], 200–500 nm thick $SiO_2$ are more typical for SWCNT-based TFTs [32,40–42].

In this paper, we report the proof of concept of an improved structure for solution-processed SWCNT-based TFTs that resolves the unusual gate leakage current. The electrodes of this bottom-gate TFT structure are inkjet-printed as an inexpensive and low-temperature alternative to traditional high temperature electrode fabrication processes. This improved structure TFT resolves the unusual gate leakage current by incorporating additional fabrication steps patterning the SWCNT layer to confine its active area to within the vicinity of the channel region. Moreover, the charge carrier mobilities and on/off ratios of our TFTs are also improved by patterning the SWCNT layer. Furthermore, we successfully created some of our TFTs using a 90 nm thick $SiO_2$ layer, which is thinner than that of similar devices reported by other groups [32,40–42]. Despite the thinness of our 90 nm thick dielectric layers, our TFTs did not suffer from the dielectric breakdown issues that have a much greater risk of occurring with thinner dielectrics [39,40]. TFTs with 300 nm thick dielectric layers were also examined, to facilitate investigating the effect of a thicker dielectric layer upon TFT performance.

Unlike traditional SWCNT TFTs [14,15], the electrodes of our TFTs are inkjet-printed as an inexpensive and low-temperature alternative to traditional electrode fabrication techniques. Our solution-processed SWCNT TFTs overcome the disadvantages of solution-processed organic TFTs in terms of both device performance and stability.

Ultimately, the improved TFTs of this paper shall in turn result in superior devices made from these TFTs, including improved sensors, logic gates, and display drivers. In the future, we will fabricate all of the layers of our SWCNT TFTs with printing technology, and also further miniaturize them, in order to develop fully printed SWCNT TFTs and circuits. Eventually, a fully solution-processed TFT structure shall be developed, so that our bottom-gate TFTs can be roll-to-roll printed on a large scale.

## 2. Materials and Methods

This paper presents four different TFTs. The critical distinctions that differentiate these four TFTs are: (i) whether its $SiO_2$ layer is 90 nm or 300 nm thick and (ii) whether it is a traditional (unpatterned) TFT with excess SWCNTs or is instead an improved (patterned) TFT without excess SWCNTs. Figure 1 overviews the step-by-step fabrication of one of our improved TFTs. The fabrication of our traditional TFTs is a nearly identical process, differing only in that the fabrication of our traditional TFTs omit the steps patterning the SWCNT active layer.

As shown in Figure 1, all of our TFTs have a bottom-gate structure, with their source (S) and drain (D) electrodes and their semiconducting active layer all situated above the dielectric layer, whereas their gate (G) electrode is located directly below the dielectric layer. Most conventional TFTs have a bottom-gate structure [32,43–45]. Bottom-gate TFTs are popular because they offer a fair balance of performance and design simplicity, whereas top-gate TFTs instead offer high performance at the cost of greater design complexity and conversely side-gate TFTs sacrifice performance for greater design simplicity [46].

Figure 1 also shows that all of our TFTs are top-contact TFTs, with their S and D electrodes situated atop their active layer. Compared to bottom-contact TFTs, which instead place their active layer above their S and D electrodes, top-contact TFTs offer a superior contact between the S and D electrodes and the active layer as well as higher on-state currents [46].

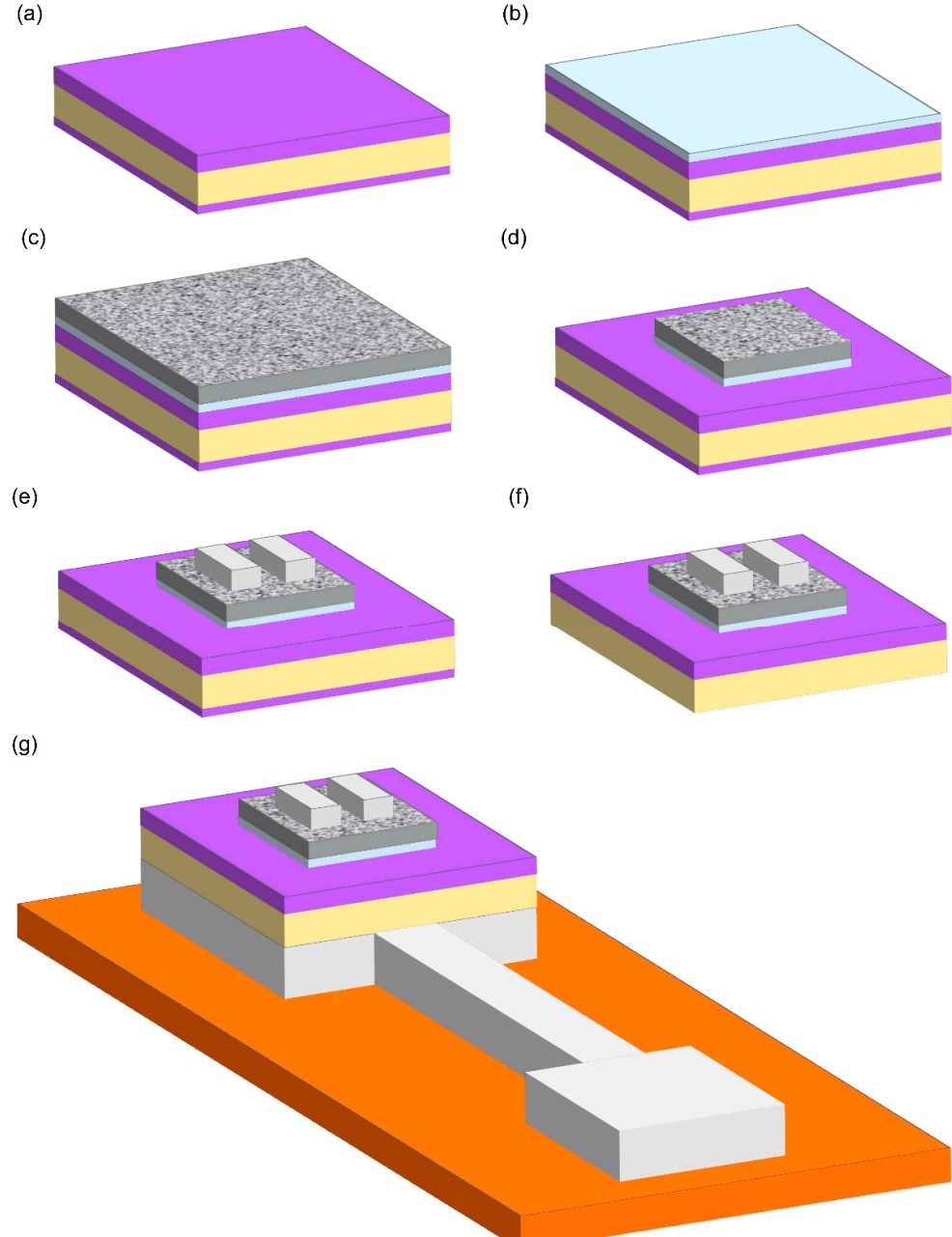

**Figure 1.** The process with which an improved TFT is fabricated. Fabrication of a traditional TFT differs only in that the steps to remove excess SWCNTs beyond the vicinity of the channel region are omitted for a traditional TFT. One of our two improved TFTs has a 90 nm thick $SiO_2$ layer, whereas the other has a 300 nm thick $SiO_2$ layer. Similarly, one of our two traditional TFTs has a 90 nm thick $SiO_2$ layer, whereas the other has a 300 nm thick $SiO_2$ layer. (**a**) Start with a <100> p-type Si (280 μm thick) substrate with a dry $SiO_2$ (90 nm or 300 nm thick) top surface and a native $SiO_2$ bottom surface. (**b**) Silanize the $SiO_2$ top surface with an APTES monolayer (~0.7 nm thick). (**c**) Deposit the SWCNT active layer atop the APTES. (**d**) In the case of an improved TFT only, pattern the SWCNT layer by removing excess SWCNTs not covered by a photolithographically patterned photoresist mask using RF plasma. Traditional TFTs differ from improved TFTs only by omitting this step. (**e**) Print and sinter silver nanoparticle (nAg) ink defining the S and D electrodes atop the SWCNT active layer. (**f**) Scrap away the native $SiO_2$ from the bottom of the Si substrate. (**g**) Print nAg ink defining the G electrode atop a Kapton polyimide film. Smear additional nAg ink across the scrapped bottom of the Si substrate. Place the sample atop the printed G electrode nAg ink. Sinter the entire assembly.

The most crucial difference between our traditional and improved TFTs is whether excess SWCNTs beyond the vicinity of the channel area are present. In the case of the traditional TFTs, excess SWCNTs cover the entire oxide surface, extending far beyond the channel area. Conversely, in the case of the improved TFTs, the excess SWCNTs beyond the vicinity of the channel area have been removed. Another significant difference between our different TFTs is whether their $SiO_2$ layer is 90 nm thick or 300 nm thick.

All of our TFTs were built on 280 µm thick single-side polished prime grade p-type Si wafers with a 50.8 mm diameter, <100> orientation, and 0.001–0.005 Ω cm resistivity. A dry $SiO_2$ layer (with a thickness of either 90 nm or 300 nm) covers the top surface of each wafer, whereas their bottom surfaces are coated by native $SiO_2$. Each wafer is diced into more conveniently sized smaller pieces, typically quartering the wafer, prior to beginning TFT fabrication.

To achieve a uniform and dense SWCNT coverage, the given $SiO_2$ surface must first be thoroughly cleaned. To achieve such a thorough cleaning, each diced wafer piece was immersed in a 70 °C bath of Nanostrip solution for 60 min, then rinsed under running DI $H_2O$ for 3 min, and finally dried with $N_2$ gas.

Silanization of a given thoroughly cleaned $SiO_2$ surface is also necessary to obtain dense and uniform SWCNT coverage following the later SWCNT deposition step [14,47–49]. (3-Aminopropyl)triethoxysilane (APTES) has proven to be a popular material with which to successfully silanize $SiO_2$ surfaces [47–49]. As such, we also used APTES to silanize the $SiO_2$ surface of all our TFTs just prior to depositing the semiconducting SWCNTs comprising their active layers. First, baths of 50% APTES and 50% IPA (*v/v*) were created by carefully mixing 99% APTES into an equal volume of 99.9% isopropyl alcohol (IPA). Following the Nanostrip surface cleaning and subsequent rinsing and drying steps mentioned above, a given sample was then submerged into one of these room temperature APTES + IPA baths for 45 min. Only an APTES monolayer (~0.7 nm thick) must be created prior to SWCNT deposition, as additional APTES beyond such a monolayer results in less uniform SWCNT layers. In order to remove excess APTES, 99.9% IPA was squirted across each sample in three cycles of 2 min of constant squirting followed by 10 s of rest. The samples were then dried with $N_2$ gas.

Now that the samples' surfaces have been thoroughly cleaned and silanized with a monolayer of APTES, SWCNTs can be deposited upon them to form the desired dense and uniform SWCNT layers. To do so, each sample was immersed in a room temperature solution of 98% semiconducting SWCNTs from NanoIntegris Inc. (Boisbriand, Canada), for a 2.5 hr incubation. The given sample was then placed in a lightly agitated bath of 99.9% IPA for 5 min, then sprayed with additional 99.9% IPA, rinsed under running DI $H_2O$ for 3 min, and finally dried with $N_2$ gas. These post-deposition steps remove any residual sodium dodecyl sulfate (SDS) remaining on the sample surfaces following incubation in the SWCNT solution. To ensure that all residual SDS has been removed from the surface, the given sample was then annealed for 1 h in a 15 µTorr and 200 °C vacuum chamber. Figure 2 presents field electron scanning electron microscope (FE-SEM) micrographs of deposited SWCNTs adhered to our $SiO_2$ surfaces following all of the fabrication processes discussed thus far. Figure 2a,b present the SWCNTs adhered to the 90 nm and 300 nm thick $SiO_2$ surfaces, respectively. The dense SWCNT coverage depicted in Figure 2a,b following silanization of our $SiO_2$ surfaces with an APTES monolayer agrees with the work of Wang et al. [14]. Specifically, Wang et al. explicitly show that a comparably dense SWCNT coverage is achieved following silanization of an $SiO_2$ surface with an APTES monolayer, whereas forgoing silanization instead results in a far sparser SWCNT coverage. The dense SWCNT coverage achieved thanks to silanization yields active layers suitable for TFTs, yielding TFTs with high on/off ratios and high mobilities thanks to its numerous SWCNT-to-SWCNT junctions. Conversely, the far sparser SWCNT coverage that results without silanization would be unsuitable as an active layer for TFTs and would not result in a functioning TFT.

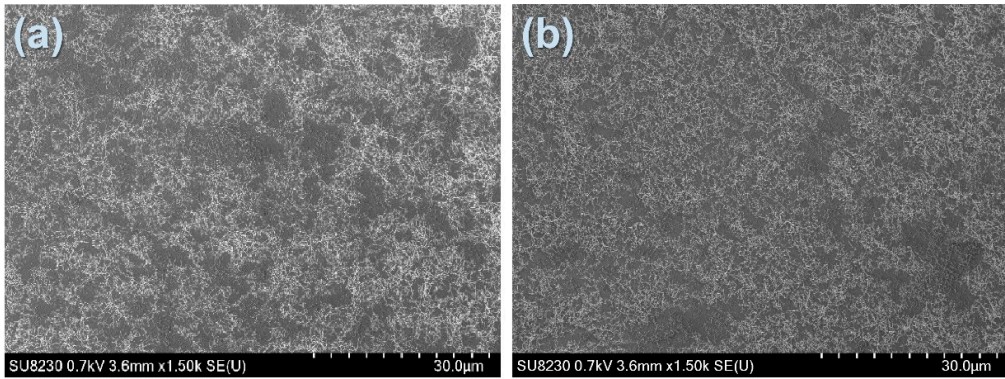

**Figure 2.** FE-SEM micrographs of deposited SWCNTs atop cleaned and APTES-silanized $SiO_2$ surfaces that are (**a**) 90 nm thick and (**b**) 300 nm thick.

Each sample was then diced into even smaller pieces, which were then used to make the traditional and improved TFTs. It is important to use pieces from the same sample that was processed as described above, in order to minimize the variation in SWCNT density between these TFTs. Variations in SWCNT density and mean SWCNT length alike should be avoided as much as possible since such variations can significantly affect the performance of TFTs [15]. As such, the two TFTs in this paper with 90 nm thick $SiO_2$ layers were both made using the surface depicted in Figure 2a. Similarly, the other two TFTs with 300 nm thick $SiO_2$ layers were both made using the surface depicted in Figure 2b.

Pairs of electrodes were then inkjet printed atop the SWCNT layers of the four different diced pieces using a conductive silver nanoparticle (nAg) ink, creating the S and D electrodes for all four TFTs. These pieces were then placed in a 150 °C oven for 30 min, to sinter the ink. Profilometry later estimated the thickness of these sintered S and D electrodes as being 100 μm thick.

Notably, the dimensions of the S and D electrodes of a given TFT are crucial as these electrode dimensions in turn define the dimensions of the TFT's semiconducting channel region. Specifically, the length of the S and D electrodes defines the channel width (W) and the edge-to-edge separation between the S and D electrodes defines the channel length (L). Table 1 summarizes all of the critical dimensions of each of our TFTs following the fabrication steps mentioned thus far. Notably, the four TFTs presented in this paper are an improved version of an earlier TFT design that had longer S and D electrodes [50]. The S and D electrodes of the four TFTs presented in this paper were made to be half as long as the S and D electrodes of these two earlier TFTs, to help avoid rapid dielectric breakdowns. For comparative purposes, Table 1 also includes the critical dimensions of these two earlier TFTs.

**Table 1.** The critical dimensions of each of our TFTs.

| TFT | SWCNT Layer Patterned? | $SiO_2$ Thickness $t_{ox}$ (nm) | Length of S and D Electrodes (mm) | Width of S and D Electrodes (μm) | Separation between S and D Electrodes (μm) | Channel Width W (mm) | Channel Length L (μm) |
|---|---|---|---|---|---|---|---|
| Traditional TFT 1 * | Unpatterned | 90 | 5.4 | 840 | 320 | 5.4 | 320 |
| Improved TFT 1 * | Patterned | 300 | 6.3 | 835 | 370 | 6.3 | 370 |
| Traditional TFT 2 | Unpatterned | 90 | 2.9 | 630 | 320 | 2.9 | 320 |
| Improved TFT 2 | Patterned | 90 | 3.0 | 660 | 290 | 3.0 | 290 |
| Traditional TFT 3 | Unpatterned | 300 | 3.0 | 600 | 350 | 3.0 | 350 |
| Improved TFT 3 | Patterned | 300 | 2.9 | 610 | 330 | 2.9 | 330 |

* Earlier TFTs with longer S and D electrodes previously presented [50], shown here for comparative purposes.

Next, the active areas of the two improved TFTs were both reduced by removing the excess SWCNTs beyond the vicinities of their channel regions. Other groups successfully patterned their own SWCNT layers using RF plasma to selectively remove SWCNTs [14,51].

We follow a similar process. The S and D electrodes and the channel between them were first protected by an etch mask (comprised of 10 μm thick AZ 9245 positive photoresist deposited and patterned via standard photolithographic processes). The remaining surface area not protected by the etch mask was then subjected to ten bursts of RF plasma, removing the excess SWCNTs in those exposed regions. Each of these bursts of plasma was a 2 min long exposure to 40 sccm Ar and 10 sccm $O_2$ gases under a 70 mTorr vacuum pressure and excited by a 50 W and 13.56 MHz RF source. One minute of sample cooling between bursts is necessary to prevent significant thermal damage to the exposed surface that would otherwise occur were the surface instead exposed to a continuous 20 min exposure without cooling periods. Following the exposure to the bursts of RF plasma, the AZ 9245 photoresist etch masks atop the improved TFTs were then stripped away entirely in lightly agitated acetone baths. The traditional TFTs retained their excess SWCNTs by simply forgoing these plasma exposure steps.

The bottom native $SiO_2$ of all four sample pieces were then scrapped away. nAg ink was then smeared across these scraped regions, forming part of each TFT's G electrode. This ink was then sintered in a 150 °C oven for 30 min. The rest of the G electrodes were printed atop Kapton polyimide substrates, with a topography comprised of a probe contact pad (3 mm wide and 3 mm long) and a sample contact pad (6 mm wide and 3 mm long) connected by a thin line (610–660 μm wide and 8.0–8.5 mm long). Each sample's bottom gate was then placed atop the sample contact pad printed atop a Kapton polyimide film. It is important that this alignment be performed carefully, to avoid having nAg ink inadvertently span across the edge of the sample and form a conductive path between the G electrode and sample surface that bypasses the $SiO_2$ layer. The entire assembly was then sintered together in a 150 °C oven for 30 min. Figure 3 presents top-down photographs of all four of the TFTs presented in this paper.

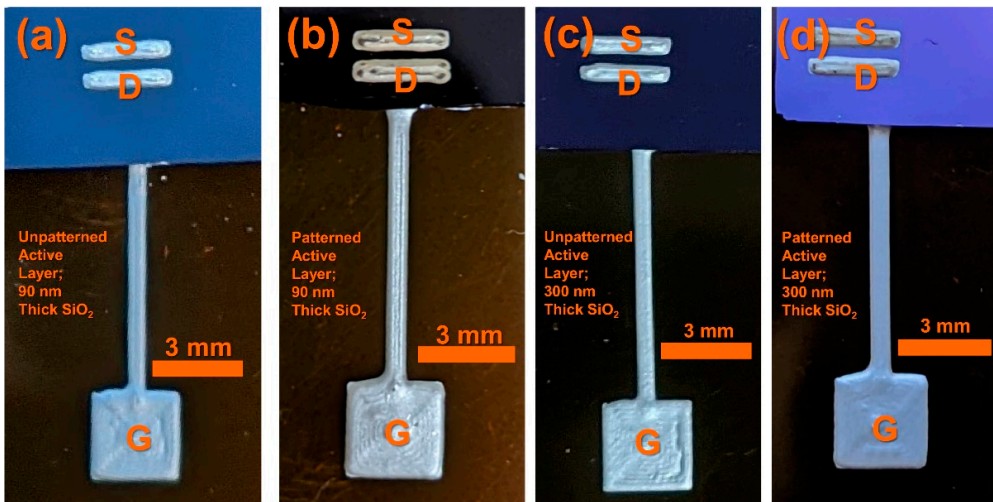

**Figure 3.** Top-down photographs of the (**a**) traditional (unpatterned) TFT with a 90 nm thick $SiO_2$, (**b**) improved (patterned) TFT with a 90 nm thick $SiO_2$, (**c**) traditional (unpatterned) TFT with a 300 nm thick $SiO_2$, and (**d**) improved (patterned) TFT with a 300 nm thick $SiO_2$.

## 3. Results

We systematically characterized the electrical properties of all four of our TFTs, using a HP 4145B Semiconductor Parameter Analyzer to measure DC drain currents ($I_D$) and gate currents ($I_G$) in response to different DC bias voltages ($V_{DS}$ and $V_{GS}$). From this characterization, we inferred the influence of patterning the semiconducting active areas and adjusting the thickness of the $SiO_2$ layers upon TFT performance. This includes determining various key TFT parameters from the response of $I_D$ to $V_{DS}$ and $V_{GS}$ and also evaluating whether the unusual $I_G$ behavior of the traditional TFTs was resolved with the improved TFTs. Multiple measurements of $I_D$ as $V_{DS}$ was swept and $V_{GS}$ was held

constant also showed that these measurements were repeatable after a six-month period. Moreover, hysteresis in the response of $I_D$ to $V_{DS}$ was observed when the direction of these $V_{DS}$ sweeps were reversed.

### 3.1. $|I_D|$/W versus $V_{DS}$ Characteristics: Confirming Ohmic Contacts and Determining on/off Ratios

Figure 4 presents the $|I_D|$/W versus $V_{DS}$ transfer characteristics of all four TFTs, as measured in the triode regime only in each case. These $|I_D|$/W versus $V_{DS}$ curves are linear for $V_{DS}$ between $-1$ V and $0$ V for all four TFTs, indicating that ohmic contacts formed between the electrodes and the SWCNTs in every case. Notably, the increase in $|I_D|$/W with $V_{DS}$ at a given $V_{GS}$ in the triode regime was greater for TFTs with patterned active regions and/or thinner $SiO_2$ layers.

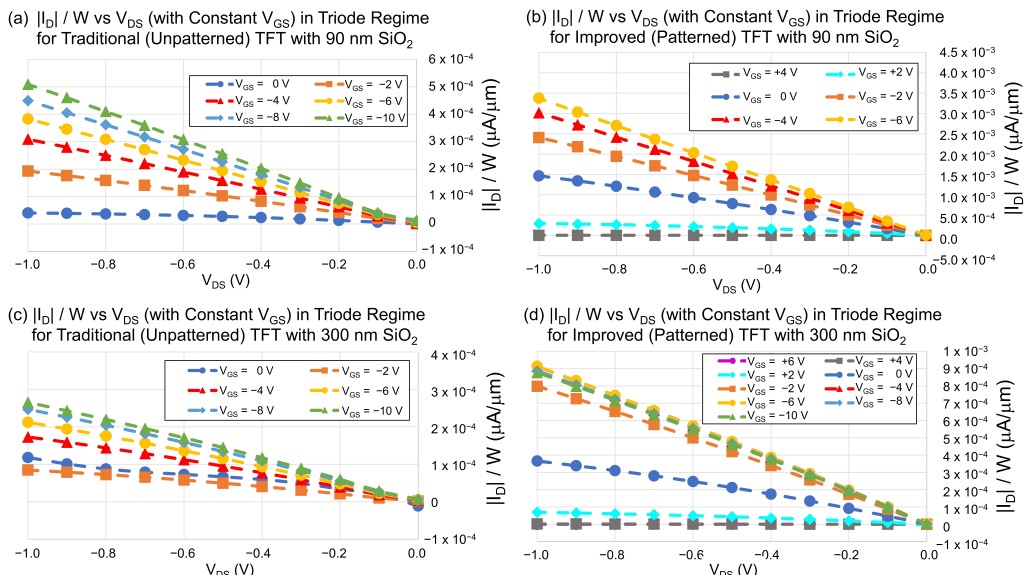

**Figure 4.** $|I_D|$/W versus $V_{DS}$ characteristics in the triode regime of the: (**a**) traditional (unpatterned) TFT with a 90 nm $SiO_2$, (**b**) improved (patterned) TFT with a 90 nm $SiO_2$, (**c**) traditional (unpatterned) TFT with a 300 nm $SiO_2$, and (**d**) improved (patterned) TFT with a 300 nm $SiO_2$.

Figure 5 presents broader $|I_D|$/W versus $V_{DS}$ characteristics for all four TFTs, as measured in the saturation regime as well as the triode regime in each case. Under more negative $V_{DS}$, the $|I_D|$/W of all four devices eventually saturate. Notably, the $|I_D|$/W versus $V_{DS}$ curves saturated with a greater magnitude for TFTs with patterned active regions and/or thinner $SiO_2$ layers.

Inspecting the $|I_D|$/W data plotted in Figure 5 allows us to determine the on/off ratio of each of our TFTs. Table 2 presents this on/off data for each TFT. As with Table 1, for comparative purposes, Table 2 also presents the on/off data of previously reported earlier TFTs that had longer S and D electrodes [50]. Notably, the on/off data for this earlier traditional TFT, with a thin 90 nm thick $SiO_2$, is unavailable as that particular TFT suffered from a rapid dielectric breakdown before sufficient $|I_D|$/W data could be acquired.

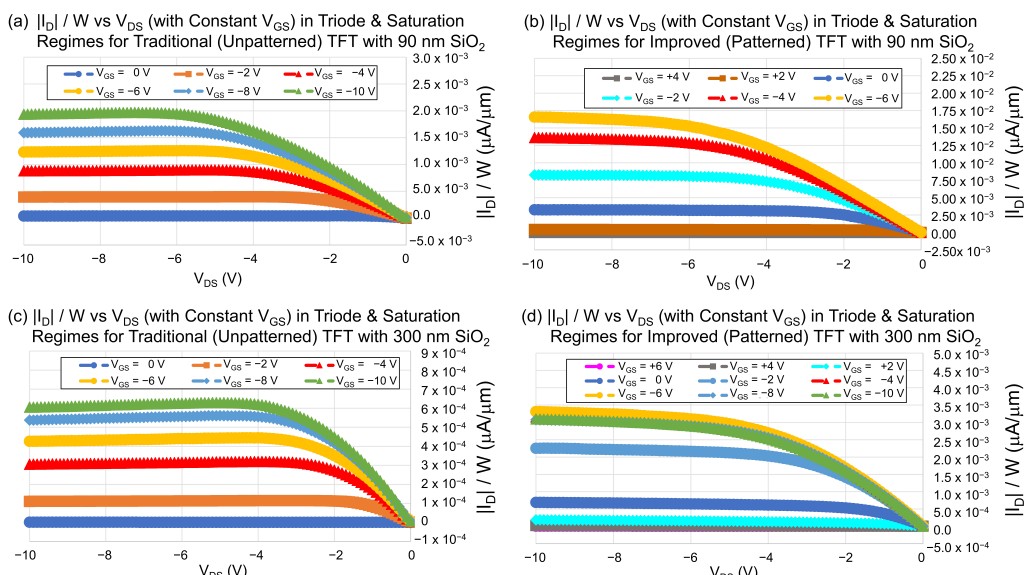

**Figure 5.** $|I_D|/W$ versus $V_{DS}$ characteristics in the triode and saturation regimes of the: (**a**) traditional (unpatterned) TFT with a 90 nm SiO$_2$, (**b**) improved (patterned) TFT with a 90 nm SiO$_2$, (**c**) traditional (unpatterned) TFT with a 300 nm SiO$_2$, and (**d**) improved (patterned) TFT with a 300 nm SiO$_2$.

**Table 2.** The on/off ratios of each TFT.

| TFT | SWCNT Layer Patterned? | SiO$_2$ Thickness $t_{ox}$ (nm) | On-State Current $|I_{ON}|/W$ ($\mu$A/$\mu$m) | Off-State Current $|I_{OFF}|/W$ ($\mu$A/$\mu$m) | On/off Ratios $I_{ON}/I_{OFF}$ (A/A) | Ratio of Improved TFT on/off Ratio/Traditional TFT on/off Ratio |
|---|---|---|---|---|---|---|
| Traditional TFT 1 * | Unpatterned | 90 | Indeterminable Due to Dielectric Breakdown | Indeterminable Due to Dielectric Breakdown | Indeterminable Due to Dielectric Breakdown | – |
| Improved TFT 1 * | Patterned | 300 | $7.97 \times 10^{-5}$ | $3.02 \times 10^{-8}$ | 2,640 | – |
| Traditional TFT 2 | Unpatterned | 90 | $1.23 \times 10^{-3}$ | $6.38 \times 10^{-8}$ | 19,280 | – |
| Improved TFT 2 | Patterned | 90 | $1.65 \times 10^{-2}$ | $6.8 \times 10^{-8}$ | 242,650 | 12.59 |
| Traditional TFT 3 | Unpatterned | 300 | $6.07 \times 10^{-4}$ | $3.33 \times 10^{-8}$ | 18,230 | – |
| Improved TFT 3 | Patterned | 300 | $3.29 \times 10^{-3}$ | $1.09 \times 10^{-7}$ | 30,180 | 1.66 |

\* Earlier TFTs with longer S and D electrodes previously presented [50], shown here for comparative purposes.

### 3.2. Repeatability and Hysteresis of $|I_D|/W$ versus $V_{DS}$ Measurements

The $|I_D|/W$ versus $V_{DS}$ transfer characteristics of the improved (patterned) TFT with a 90 nm thick SiO$_2$ layer were remeasured using multiple sweeps six-months after the initial $|I_D|/W$ versus $V_{DS}$ data presented in Figure 5b. During the intervening six-month period between these sets of measurements were recorded, the TFT was not used and was stored in a Petri dish in a room temperature environment. As before, the later set of measurements was recorded using a HP 4145B Semiconductor Parameter Analyzer. Figure 6 presents two of these later sweeps, along with the initial $|I_D|/W$ versus $V_{DS}$ data previously presented in Figure 5b. All of the $|I_D|/W$ versus $V_{DS}$ datasets shown in Figure 6 were collected with $V_{DS}$ swept as $V_{GS}$ was held constant at −6 V. The initial data (shown in Figures 5b and 6 in dark blue) was collected as $V_{DS}$ was swept from 0 V to −10 V. Six months later, this same sweep was repeated, yielding comparable $|I_D|/W$ versus $V_{DS}$ data (shown in Figure 6 in red) overlapping with the previous $|I_D|/W$ versus $V_{DS}$ dataset, thereby demonstrating the repeatability of these $|I_D|/W$ versus $V_{DS}$ measurements after six-months. Indeed, the extent of the overlap between the measured $|I_D|/W$ versus $V_{DS}$ data indicates that the TFT's characteristic parameters did not change appreciably during the intervening six-month period between the two sets of measurements. An additional $|I_D|/W$ versus $V_{DS}$ dataset, with $V_{DS}$ instead swept from −10 V to 0 V, was also acquired six-months after the initial $|I_D|/W$ versus $V_{DS}$ measurements, to investigate hysteresis. This third $|I_D|/W$ versus $V_{DS}$ dataset (shown in Figure 6 in light blue) does indeed demonstrate that

our SWCNT-based TFTs exhibit hysteresis, as together the $|I_D|/W$ versus $V_{DS}$ data from forward and reverse direction sweeps form a complete hysteresis curve.

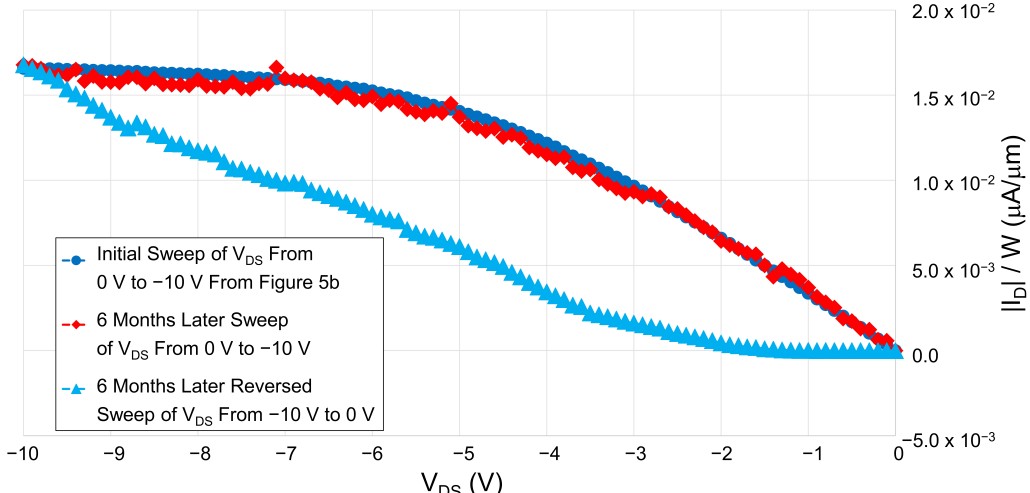

**Figure 6.** The $|I_D|/W$ versus $V_{DS}$ transfer characteristics for the improved (patterned) TFT with a 90 nm thick SiO$_2$ as $V_{GS}$ was held at $-6$ V and $V_{DS}$ was: swept from 0 V to $-10$ V as shown in Figure 5b (DARK BLUE), swept from 0 V to $-10$ V again (RED), and swept in reverse from $-10$ V to 0 V (LIGHT BLUE). The RED and LIGHT BLUE data was acquired six-months after the DARK BLUE data was acquired.

*3.3. $|I_D|/W$ versus $V_{GS}$ and $\sqrt{|I_D|/W}$ versus $V_{GS}$ Characteristics: Determining Threshold Voltages and Mobilities*

Figure 7 presents the $|I_D|/W$ versus $V_{GS}$ and $\sqrt{|I_D|/W}$ versus $V_{GS}$ transfer characteristics of all four TFTs, as measured in the saturation regime (with $V_{DS} = -8$ V) in every case. These transfer characteristics are useful as the threshold voltage ($V_{th}$) and charge carrier mobility ($\mu$) of a given TFT can be estimated from its $|I_D|/W$ versus $V_{GS}$ and $\sqrt{|I_D|/W}$ versus $V_{GS}$ curves, respectively.

The relationship between $I_D$ and $V_{GS}$ in the saturation regime is well known [52]:

$$I_D = \frac{1}{2}\frac{W}{L}\mu C_i (V_{GS} - V_{th})^2 \tag{1}$$

where $C_i$ is the capacitance of the oxide layer per unit surface area. $C_i$ is itself estimated as [52] (with $t_{ox} = 90$ nm used as an example):

$$C_i = \frac{\epsilon_{ox}}{t_{ox}} \approx \frac{(3.9)\left(8.854 \times 10^{-14} \text{ F/cm}\right)}{9 \times 10^{-6} \text{ cm}} \approx 3.84 \times 10^{-8} \frac{\text{F}}{\text{cm}^2} \tag{2}$$

The relationship between $\sqrt{|I_D|}$ and $V_{GS}$ in the saturation regime follows by simply taking the square root of Equation (1) [52]:

$$\sqrt{|I_D|} = \sqrt{\frac{1}{2}\frac{W}{L}\mu C_i}\,V_{GS} - \sqrt{\frac{1}{2}\frac{W}{L}\mu C_i}\,V_{th} \tag{3}$$

Using Equation (3) in conjunction with the $\sqrt{|I_D|/W}$ versus $V_{GS}$ curves of Figure 7, the μ of a given TFT can be estimated via:

$$\mu = 2\frac{L}{W}\frac{1}{C_i}\left(\frac{\Delta\sqrt{|I_D|}}{\Delta V_{GS}}\right)^2 \tag{4}$$

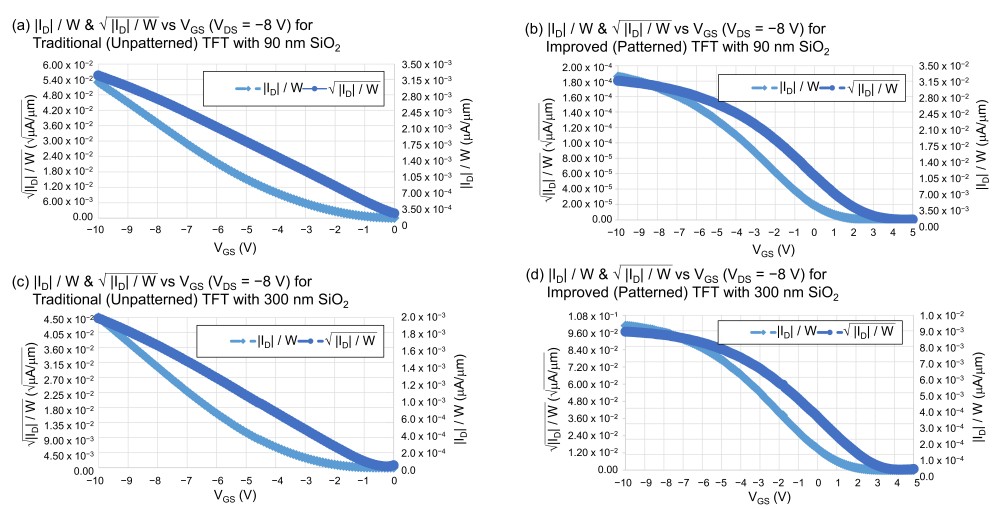

**Figure 7.** The $|I_D|/W$ versus $V_{GS}$ (LIGHT BLUE) and $\sqrt{|I_D|/W}$ versus $V_{GS}$ (DARK BLUE) transfer characteristics, measured in the saturation regime (with $V_{DS} = -8$ V), for the: (**a**) traditional (unpatterned) TFT with 90 nm SiO$_2$, (**b**) improved (patterned) TFT with 90 nm SiO$_2$, (**c**) traditional (unpatterned) TFT with 300 nm SiO$_2$, and (**d**) improved (patterned) TFT with 300 nm SiO$_2$.

Table 3 presents the $V_{th}$ and μ of each TFT, as estimated from the data plotted in Figure 7. $V_{th}$ was determined via visual inspection of the $|I_D|/W$ versus $V_{GS}$ data in Figure 7, whereas μ was determined by plugging the $\sqrt{|I_D|/W}$ versus $V_{GS}$ data from Figure 7 into Equations (1)–(4). As with Tables 1 and 2, for comparative purposes, Table 3 also presents the $V_{th}$ and μ of previously reported earlier TFTs that had longer S and D electrodes [50]. Notably, the $V_{th}$ and μ data for this earlier traditional TFT, with a thin 90 nm thick SiO$_2$, is unavailable as that particular TFT suffered from a rapid dielectric breakdown before sufficient $I_D$ data could be acquired.

**Table 3.** The threshold voltage and charge carrier mobility of each TFT.

| TFT | SWCNT Layer Patterned? | SiO$_2$ Thickness $t_{ox}$ (nm) | Threshold Voltage $V_{th}$ (V) | Charge Carrier Mobility μ (cm$^2$ V$^{-1}$ s$^{-1}$) | Ratio of Improved TFT μ/Traditional TFT μ |
|---|---|---|---|---|---|
| Traditional TFT 1 * | Unpatterned | 90 | Indeterminable Due to Dielectric Breakdown | Indeterminable Due to Dielectric Breakdown | – |
| Improved TFT 1 * | Patterned | 300 | −1.5 | 0.3 | – |
| Traditional TFT 2 | Unpatterned | 90 | +1.96 | 0.51 | – |
| Improved TFT 2 | Patterned | 90 | +2.63 | 7.82 | 15.3 |
| Traditional TFT 3 | Unpatterned | 300 | −1.73 | 1.48 | – |
| Improved TFT 3 | Patterned | 300 | +3.43 | 8.10 | 5.47 |

* Earlier TFTs with longer S and D electrodes previously presented [50], shown here for comparative purposes.

### 3.4. $G_m/W$ versus $V_{GS}$ Characteristics: Transconductance

Section 3.1 presented plots of $|I_D|/W$ versus $V_{GS}$, as $V_{DS}$ is held constant, for each TFT as Figure 7. Having such plots, we can proceed to determine the transconductance ($G_m = \Delta I_D/\Delta V_{GS}$) of each TFT, again normalized to the channel width W. Figure 8 presents plots of $G_m/W$ versus $V_{GS}$ as $V_{DS}$ is held constant.

The $G_m/W$ data plotted in Figure 8 clearly shows that, for the same applied $V_{GS}$ and $V_{DS}$, the improved TFTs with small patterned active areas will exhibit a larger $G_m/W$ in the on-state than the traditional TFTs with large unpatterned active areas. Moreover, for the same applied $V_{GS}$ and $V_{DS}$, TFTs with thin 90 nm $SiO_2$ layers had larger $G_m/W$ in the on-state than TFTs with thicker 300 nm $SiO_2$ layers. The patterning of the active layer had a more substantial effect increasing $G_m/W$ than thinning the $SiO_2$ layer.

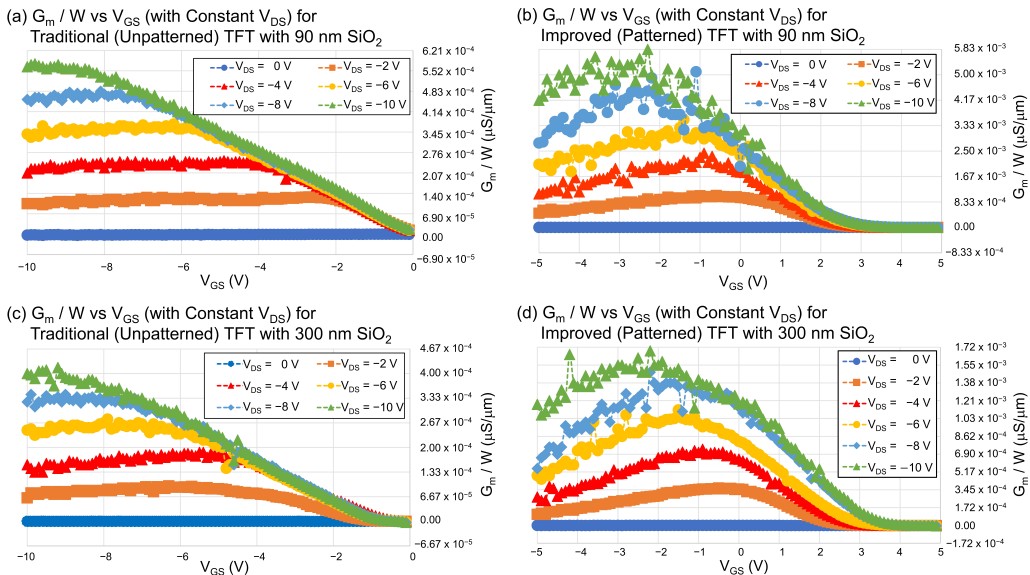

**Figure 8.** $G_m/W$ versus $V_{GS}$ characteristics (with constant $V_{DS}$) measured for the: (**a**) traditional (unpatterned) TFT with 90 nm $SiO_2$, (**b**) improved (patterned) TFT with 90 nm $SiO_2$, (**c**) traditional (unpatterned) TFT with 300 nm $SiO_2$, and (**d**) improved (patterned) TFT with 300 nm $SiO_2$.

### 3.5. $|I_G|$ versus $V_{GS}$ Characteristics: Resolving Unusual Gate Leakage Current Behavior

To gauge the efficacy of SWCNT patterning in resolving unusual $I_G$ behavior, we also measured the $|I_G|$ versus $V_{GS}$ characteristics of every TFT. Figure 9 presents $|I_G|$ versus $V_{GS}$ for each TFT.

Table 4 presents the $|I_G|$ of each TFT (as measured with $V_{DS} = 0$ V and $V_{GS} = -5$ V) as estimated from the data plotted in Figure 9. As with Tables 1–3, for comparative purposes, Table 4 also presents the $|I_G|$ of previously reported earlier TFTs that had longer S and D electrodes [50]. Notably, the $|I_G|$ data for this earlier traditional TFT, with a thin 90 nm thick $SiO_2$, is available as its rapid dielectric breakdown occurred after the relevant $|I_G|$ data had already been acquired.

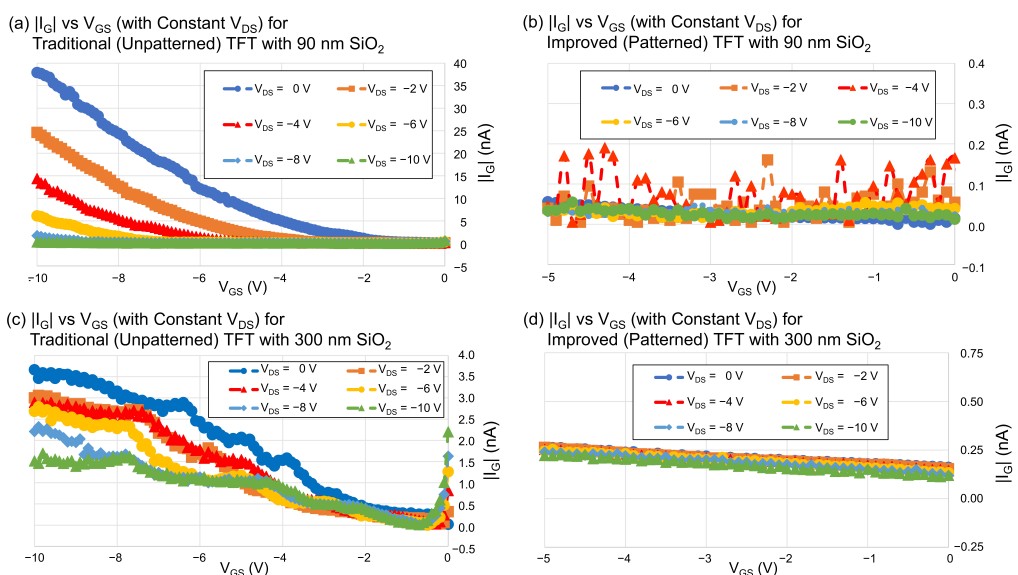

**Figure 9.** $|I_G|$ versus $V_{GS}$ characteristics (with constant $V_{DS}$) measured for the: (**a**) traditional (unpatterned) TFT with 90 nm $SiO_2$, (**b**) improved (patterned) TFT with 90 nm $SiO_2$, (**c**) traditional (unpatterned) TFT with 300 nm $SiO_2$, and (**d**) improved (patterned) TFT with 300 nm $SiO_2$.

**Table 4.** Comparing the gate currents of traditional and improved TFTs.

| TFT | SWCNT Layer Patterned? | SiO$_2$ Thickness $t_{ox}$ (nm) | Gate Current $|I_G|$ (with $V_{DS}$ = 0 V) | Ratio of Improved TFT $|I_G|$/Traditional TFT $|I_G|$ |
|---|---|---|---|---|
| Traditional TFT 1 * | Unpatterned | 90 | 3.5 µA | – |
| Improved TFT 1 * | Patterned | 300 | 420 pA | 8,330 |
| Traditional TFT 2 | Unpatterned | 90 | 10 nA | – |
| Improved TFT 2 | Patterned | 90 | 60 pA | 170 |
| Traditional TFT 3 | Unpatterned | 300 | 2.1 nA | – |
| Improved TFT 3 | Patterned | 300 | 260 pA | 8 |

* Earlier TFTs with longer S and D electrodes previously presented [50], shown here for comparative purposes.

## 4. Discussion

The traditional TFTs, with large unpatterned SWCNT-based active layers, exhibited unusual $I_G$ behavior defying TFT expectations, with large magnitudes that varied with the applied $V_{GS}$ and $V_{DS}$. These unusual traditional TFT $|I_G|$ versus $V_{GS}$ characteristics indicate that electrical charge carriers were actively transported through the $SiO_2$ layer situated between the G electrode and the D and S electrodes at a negative $V_{GS}$, defying the expected $I_G$ for a TFT. This is the case whether the $SiO_2$ layer is 90 nm or 300 nm thick. This transportation of electrical charge carriers through the $SiO_2$ layer was significantly reduced in the case of the improved TFTs, due to having patterned their active layers to remove excess SWCNTs beyond the vicinities of their channel regions. Comparing the two TFTs with 300 nm thick $SiO_2$ layers, the improved TFT had a $I_G$ that was 8 times lower than that of the traditional TFT with the same dimensions (aside from the unpatterned active layer area) in the on-state. In the case of the other two TFTs with 90 nm thick $SiO_2$ layers, the improved TFT had a $I_G$ that was 170 times lower than that of the traditional TFT with the same dimensions (aside from the unpatterned active layer area) in the on-state. Moreover, in both the 90 nm and 300 nm thick $SiO_2$ cases, the variation of $I_G$ with applied $V_{DS}$ and $V_{GS}$ in the improved TFTs was greatly reduced as compared to the traditional TFTs. Indeed, the variation in $I_G$ with $V_{GS}$ and $V_{DS}$ ranged from exceptionally weak to completely negligible in the case of the improved TFTs, whereas this variation was rather strong in the case of the traditional TFTs. The $I_G$ of the improved TFTs therefore satisfied the expectations for

a TFT, having resolved the unusual behavior exhibited by the traditional TFTs with large unpatterned SWCNT layers. That the unusual gate leakage current behavior was exhibited by the TFTs with large unpatterned active areas but not by the TFTs with smaller patterned active areas agrees with what is now known about the source of the unusual gate leakage current behavior. Specifically, that the unusual gate leakage current behavior is the result of the conduction of electrons through defects and trap sites inside the dielectric layer, with larger active layer areas encompassing a greater volume of such defects and trap sites, thereby leading to the unusual gate leakage current behavior in TFTs with large active layers [32].

Comparing the $|I_D|/W$ characteristics of the traditional and improved TFTs also demonstrated that patterning the SWCNT-based active layers moreover enhanced the on/off ratios, $\mu$, and $G_m/W$ of the improved TFTs as compared to the traditional TFTs with large unpatterned active layers. Comparing the two TFTs with 300 nm thick $SiO_2$ layers, the improved TFT had an on/off ratio nearly 1.7 times greater and a $\mu$ nearly 5.5 times greater than that of the traditional TFT with the same dimensions (aside from the unpatterned active layer area) in the on-state. In the case of the other two TFTs with 90 nm thick $SiO_2$ layers, the improved TFT had an on/off ratio nearly 12.6 times greater and a $\mu$ nearly 15.3 times greater than that of the traditional TFT with the same dimensions (aside from the unpatterned active layer area) in the on-state. The mobilities of the two new improved TFTs presented in this paper were 7.82 $cm^2\ V^{-1}\ s^{-1}$ and 8.10 $cm^2\ V^{-1}\ s^{-1}$. These mobilities are comparable to the 10 $cm^2\ V^{-1}\ s^{-1}$ achieved by Wang et al. when fabricating TFTs with an average SWCNT length of 1.0 $\mu$m [14]. Moreover, the on/off ratios of our TFTs are comparable to the on/off ratios on the order of $10^4$ A/A reported by Wang et al. [14]. Similarly, the $G_m/W$ values between $10^{-4}$–$10^{-2}$ $\mu$S/$\mu$m exhibited by our improved TFTs are not dissimilar to the $10^{-3}$–1 $\mu$S/$\mu$m reported by Wang et al. [14]. Conversely, our two traditional TFTs performed worse than our two improved TFTs and the TFTs reported by Wang et al. with respect to each of these performance metrics. Notably, Wang et al. achieved TFTs with a higher $\mu$ of 52 $cm^2\ V^{-1}\ s^{-1}$ using SWCNTs with a longer 1.7 $\mu$m average length, thereby demonstrating the strong correlation between $\mu$ and average SWCNT length, due to the increased number of SWCNT-to-SWCNT junctions resulting from a greater SWCNT length [14]. As the SWCNTs we purchased from NanoIntegris Inc. to create our active layers have an average length of approximately 1 $\mu$m, we would expect to achieve a $\mu$ more comparable to that achieved by Wang et al. using 1 $\mu$m long SWCNTs rather than the greater $\mu$ they achieved using longer 1.7 $\mu$m SWCNTs, as is indeed the case with our improved TFTs. To achieve TFTs with a greater on/off ratio and $\mu$ in the future, we can further refine our fabrication process to produce an even denser coverage of even longer SWCNTs.

The $V_{th}$ of our TFTs varied considerably, ranging from $-1.73$ V to $+3.43$ V. Numerous factors contribute to the variability in $V_{th}$ between otherwise similar FETs, including random dopant fluctuation, various oxide charges, the Body Effect, etc. [15]. Moreover, in the case of FETs with SWCNT-based active layers in particular, various SWCNT/SWCNT, SWCNT/substrate, SWCNT/oxide, and SWCNT/electrode interactions also contribute to variations in $V_{th}$ between FETs [53]. Variations in the uniformity and density of SWCNT films between different devices can also contribute to this variation in $V_{th}$. Consequentially, the $V_{th}$ of SWCNT-based TFTs can vary considerably and moreover can be difficult to reliably tune, which hinders optimization of circuits utilizing SWCNT-based TFTs. Regardless, some groups have successfully controlled and finetuned the $V_{th}$ of their SWCNT-based TFTs. For example, Wang et al. reported a facile method to finetune the $V_{th}$ of SWCNT-based TFTs by controllably doping SWCNTs using 1 *H*-benzoimidazole derivatives processed via either solution coating or vacuum deposition [54]. This n-doping of SWCNTs changes the charge carrier density within the doped SWCNT active layer, shifting its Fermi level towards the edge of its conduction band, reducing the thickness of its Schottky barrier for electron injection, and thus shifting the $V_{th}$ of the resulting TFTs towards a more negative voltage. Moreover, Wang et al. demonstrated that the controlled doping of their SWCNTs

can reliably achieve the desired $V_{th}$ with a small variation between TFTs. However, this doping of SWCNTs also affects $\mu$, requiring that care be taken to ensure that $\mu$ is not reduced too much as a consequence of this doping. As such a solution coating process to dope SWCNTs is compatible with the fabrication of our own SWCNT-based TFTs as discussed in this paper, we shall consider incorporating such an additional SWCNT doping step into the fabrication of subsequent TFTs in the future, to finetune the $V_{th}$ of those TFTs and reduce the variation in $V_{th}$ between those TFTs.

Six months after the initial measurements of $I_D$ as $V_{DS}$ was swept and $V_{GS}$ was held constant, these measurements were repeated with overlapping $|I_D|/W$ versus $V_{DS}$ data as the result, thereby demonstrating that such measurements for our TFTs were repeatable after a six-month period. Moreover, by reversing the direction of these $V_{DS}$ sweeps, we confirmed that the $|I_D|/W$ versus $V_{DS}$ characteristics of our TFTs exhibit hysteresis. This hysteresis is not unexpected, as it is known that the interface of an organic semiconducting layer (i.e., the deposited SWCNTs) with an inorganic oxide (i.e., the $SiO_2$ layer) features a high density of traps [8]. As previously discussed, a high density of traps inside the $SiO_2$ layer is one of the main causes of the unusual gate leakage current behavior seen with our traditional TFTs [32]. These high density traps also result in the hysteresis seen with TFTs built with such an interface [8]. Although the unusual gate leakage current behavior of our improved TFTs was resolved by patterning their active layers, hysteresis in the $I_D$ versus $V_{DS}$ characteristics of the improved TFTs was still observed regardless.

Previously, we presented work in which a traditional TFT with unpatterned SWCNT-based active layers and a 90 nm thick $SiO_2$ layer suffered from a rapid dielectric breakdown, whereas a similar improved TFT with a patterned SWCNT layer and a thicker 300 nm thick $SiO_2$ did not suffer from a rapid dielectric breakdown [50]. Prior to those two earlier TFTs, we had fabricated eleven additional TFTs that had all suffered from dielectric breakdown. It is only after patterning the SWCNT active layer and increasing the $SiO_2$ thickness that we were finally able to successfully fabricate TFTs that did not suffer from a rapid dielectric breakdown. In this paper, we present four additional TFTs differing from each other with regard to whether their SWCNT-based active layer is patterned and whether their $SiO_2$ layer is 90 nm or 300 nm thick. These four TFTs also differed from all of our previous TFTs, including those in [50], as the top electrodes of those earlier TFTs were twice as long as the top electrodes of the four later TFTs presented in this paper. Notably, none of the four TFTs presented in this paper suffered from a rapid dielectric breakdown. This is the case because reducing the top electrode length, reducing the semiconducting active area, and increasing the thickness of the $SiO_2$ layer are all steps that make it less likely that strong induced electric fields or a build-up of defects and ejected electrode ions during percolation will form conductive bridges bridging across the $SiO_2$ layer and causing a rapid dielectric breakdown [39,40]. Having twelve TFTs that all suffer from a rapid dielectric breakdown prior to these refinements be followed by five TFTs that did not suffer from a rapid dielectric breakdown strongly suggests that these three steps do indeed help avoid a rapid dielectric breakdown.

As well as making rapid dielectric breakdowns less likely, changing the thickness of the $SiO_2$ layer also has a significant impact upon TFT performance in several additional ways. This is to be expected, as Equation (2) shows that $C_i$ is inversely proportional to $t_{ox}$ and $C_i$ is in turn a key parameter for $\mu$ and $I_D$ as shown in Equations (4) and (1), respectively. As $\mu$ is inversely proportional to $C_i$, it is therefore directly proportional to $t_{ox}$. Indeed, examining Table 3 shows that $\mu$ increases with $t_{ox}$ as expected. Conversely, Figure 5 and Table 2 shows that the on/off ratio will increase as $t_{ox}$ is decreased. All of this behavior meets the expectations for a TFT. Examining the traditional TFTs with unusual $I_G$ behavior, it is observed that the magnitude of the unusually large $I_G$ in the on-state is even larger when $t_{ox}$ is thinner, as shown in Figure 9 and Table 4. This makes sense, as it is known that the unusual $I_G$ depends upon the vertical electric field applied across the $SiO_2$ layer and that the vertical electric field moreover varies with $t_{ox}$ [32]. Conversely, in the case of the

improved TFTs, where $I_G$ is expected to be exceedingly small and invariant with $V_{DS}$ and $V_{GS}$, $t_{ox}$ has much less of an effect upon $I_G$, as is observed in Figure 9 and Table 4.

## 5. Conclusions

We presented the proof of concept of an improved structure for solution-processed SWCNT-based TFTs. Most crucially, this improved TFT structure uses RF plasma to pattern the SWCNT-based active layers, removing excess SWCNTs beyond the vicinity of the channel region. The improvements in TFT performance yielded by patterning the SWCNT active layer include resolving unusual $I_G$ behavior, enhancing on/off ratios and $\mu$, and helping to avoid rapid dielectric breakdowns. Clearly, these advantages from patterning the SWCNT active layer demonstrate that it should be considered a crucial step that should not be omitted during fabrication of SWCNT-based TFTs. It was also found that increasing the $SiO_2$ thickness and reducing the top electrode length also helped prevent rapid dielectric breakdowns.

The electrodes and the active layers of the TFTs presented in this paper were deposited using low-temperature solution-processed techniques. In the future, the active and dielectric layers will be printed and the TFT shall be further miniaturized, to produce entirely printed high performance SWCNT-based TFTs and circuits.

**Author Contributions:** Conceptualization, S.F.R. and S.B.; methodology, S.F.R., S.B. and B.R.; validation, S.F.R. and S.B.; formal analysis, S.F.R. and S.B.; investigation, S.F.R. and S.B.; resources, S.B. and P.-L.G.-L.; data curation, S.F.R.; writing—original draft preparation, S.F.R.; writing—review and editing, S.F.R. and S.B.; visualization, S.F.R. and S.B.; supervision, S.B. and P.-L.G.-L.; project administration, S.B.; funding acquisition, S.B. All authors have read and agreed to the published version of the manuscript.

**Funding:** This research was funded by a National Sciences and Engineering Research Council of Canada (NSERC) Alliance grant in partnership with iMD Research, Object Research Systems (ORS) Inc., and MEDTEQ, with grant numbers G255618 NSERC ALLRP 556903-20, G255665 Object RS ALLRP 556903-20, G255666 iMD Res ALLRP 556903-20, and G256102 MEDTEQ ALLRP 556903-20.

**Institutional Review Board Statement:** Not applicable.

**Informed Consent Statement:** Not applicable.

**Data Availability Statement:** Not applicable.

**Acknowledgments:** The authors thank our colleagues Yiwen Chen, Seyedfakhreddin Nabavi, Zhao Lu, Jun Li, Stéphanie Bessette, Florence Paray, Christophe Clément, Ricardo Izquierdo, and Andy Shih for their assistance.

**Conflicts of Interest:** The authors declare no conflict of interest.

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
