# Peer review of "Resolving the Unusual Gate Leakage Currents of Thin-Film Transistors with Single-Walled Carbon-Nanotube-Based Active Layers"

_electronics, doi:10.3390/electronics11223719_

Round 1
Reviewer 1 Report
The manuscript reports the study of resolving the unusual gate leakage currents of TFT with SWCNT active layers. The results are interesting even a similar work have published. However, several comments have to address as following:
1. What is the equipment for current-voltage characteristics? The resolution of equipment is important.
2. The transconductance characteristics (Ids/Vgs) are required.
3. How about the dynamic response?
4. How about the mobility of the active layers?
5. How about the on/off current ratio (Ion/Ioff)? And, please make a compare to the published TFT with SWCNT network active layer.
Therefore, I recommend it as major revision to publish.
Author Response
ANSWER TO QUESTION 1:
The equipment used to collect this data was a HP 4145B Semiconductor Parameter Analyzer. The revised manuscript now explicitly states this, in Lines 285 and 334.
ANSWER TO QUESTION 2:
A new subsection (Section 3.4 spanning Lines 401-415) discussing Gm/W has been added to the revised manuscript. This new subsection includes a new figure presenting the Gm/W vs VGS characteristics of each TFT at different VDS values. This new figure shows that, for the same VDS and VGS, TFTs with smaller patterned active layers or thinning SiO2 layers had a larger Gm/W in the on-state. Patterning the active layer had a much greater impact increasing Gm/W than thinning the SiO2 layer. A discussion of Gm/W has also been added to the Discussion section of the revised manuscript, in the paragraph spanning Lines 464-491.
ANSWER TO QUESTION 3:
We did not characterize the dynamic response of our TFTs and do not have the time to do so in the limited timeframe provided for revising our manuscript. The intent of our paper was to explore how the unusual gate leakage current and rapid dielectric breakdown issues were resolved by the various refinements presented in this paper. The DC ID and IG in response to DC VDS and VGS were measured to explore these issues and we computed the applicable characteristics from that DC current and voltage data. Additional data acquisition and characterization, such as the dynamic response, is outside of the scope of this paper.
ANSWER TO QUESTION 4:
The mobility of each TFT is already listed in Table 3, which spans Lines 398-400 of the revised manuscipt. These mobilities were computed using ID flowing from source to drain through the active layer. They are the mobilities of the active layer. These tabulated mobilities are discussed at length in Section 3.3 (Lines 358-400) and Section 4 (Lines 464-491).
ANSWER TO QUESTION 5:
The Ion/Ioff ratio of each TFT is already listed in Table 2, which spans Lines 325-327 of the revised manuscript. These tabulated Ion/Ioff ratios are discussed at length in Section 3.1 (Lines 319-327) and Section 4 (Lines 464-491). Additional material discussing Vth and comparing our TFTs to those fabricated by Wang et al. in numerous ways have been added to the Discussion section (Lines 464-515). The Introduction has also been expanded with further discussion concerning SWCNT based TFTs (Lines 54-83).
Reviewer 2 Report
The title of the manuscript is "Resolving the Unusual Gate Leakage Currents of Thin Film Transistors with Single-Walled Carbon Nanotube Based Active Layers". However, the content does not follow the title entirely. Although there are some interesting points in the manuscript, the following concerns should be considered before further consideration by the journal.
1. Authors claimed "avoid dielectric breakdown" as one of the merits of the proposed technology. However, no evidence is shown in the manuscript. Authors should show the related reliability experimental results to verify the claim.
2.What's the meaning of Fig.2? It doesn't include enough data, with only the surface contour to be shown.
3.Authors should show the technology process, corresponding to the contents in SECTION II.
4.For the sample to be measured in 6 months, what's the environment? Authors should list the related parameters of the sample within 6 months to show whether there is variation of the device characteristic.
5. It is expected that the authors should propose a physics explanation on the mechanism of the improvement. Why the technology can improve the property? However, there isn't any discussion on it.
Author Response
ANSWER TO QUESTION 1:
The original manuscript discussed how one of the earlier TFTs presented in our ISCAS 2022 conference paper suffered from a rapid dielectric breakdown, whereas all five later TFTs fabricated after refining our design and fabrication processes (to use shorter top electrodes, thinner SiO2 layers, and/or patterned SWCNT active layers) did not suffer from a rapid dielectric breakdown. Admittedly, presenting only one failure compared to five subsequent successes constitutes a small sample size. Prior to the TFTs presented in our ISCAS 2022 conference paper, we had fabricated eleven even earlier TFTs that did not incorporate any of the aforementioned and all suffered a rapid dielectric breakdown. Having twelve TFTs that all suffered from a rapid dielectric breakdown prior to this refinements and then five TFTs that all did not suffer from a rapid dielectric breakdown after these refinements strongly suggests that these refinement prevent a rapid dielectric breakdown. Additional commentary mentioning these additional, earlier failures to strength our argument has been added to the revised manuscript’s Discussion section (Lines 529-549).
ANSWER TO QUESTION 2:
Figure 2 are SEM micrographs simply showing the coverage of deposited SWCNTs atop our functionalized surfaces. It is meant to show only that. However, we have added additional commentary to Section 2 further discussing the relevance of Figure 2 (Lines 206-216). Specifically, we discuss how the SWCNT density depicted in Figure 2 agrees with that shown in:
Wang, C.; Zhang, J.; Ryu, K.; Badmaev, A.; De Arco, L. G.; Zhou, C. A wafer-scale fabrication of separated carbon nanotube thin-film transistors for display application. Nano Lett. 2009, 9(12): 4285–4291.
Wang et al.’s paper shows that silanization of a SiO2 surface yields a SWCNT density comparable to that of our Figure 2, which is suitable as an active layer for TFTs. Conversely, Wang et al.’s paper also shows that omitting silanization results in a far sparser SWCNT coverage that is unsuitable as an active layer for TFTs. Had we taken SEM images of SWCNTs sparsely deposited atop an un-silanized SiO2 surface, we could have also presented it to compare. Instead, we now discuss Wang et al.’s paper, which was highly influential for us. Moreover, additional commentary discussing the effect of SWCNT density has been added to the Introduction section (Lines 54-83).
ANSWER TO QUESTION 3:
We have changed Figure 1 (Line 144) so that instead of merely showing the cross-section of a complete TFT it presents a 3D view of a TFT at various stages of its fabrication, thereby illustrating the fabrication process in a step-by-step manner.
ANSWER TO QUESTION 4:
The TFT was unused and stored in a Petri dish at room temperature during this intervening six month period. The new measurements of ID vs VDS show repeatability by reproducing essentially the same data six months later (with additional data to demonstrate hysteresis). Indeed, despite the six month interim, the curves virtually overlap and the ID versus VDS characteristic is the same. Therefore, the parameters determined from the ID versus VDS characteristic would clearly also be the same. Sentences explicitly stating all of this has been added to Section 3.2 (Lines 328-357).
ANSWER TO QUESTION 5:
The physics behind why the unusual gate leakage behaviour occurs with large unpatterned active layers but not with small patterned active layers was discussed in the Introduction (Lines 95-102). Namely, that the large active area encompasses a greater volume of trap sites within the oxide. Additional sentence to reiterate this explanation will be added to the Discussion section (Lines 456-463).
Reviewer 3 Report
In general, your whole work is good, but couple of issues must be explained or added into illustration. The English quality must be checked again.
In the technical issues, you can comment them or add some supplement.
1. According to the article, the gate leakage currents of TFTs with SWCNT based active layers are obviously improved. It’s great. In your measurement results, as shown in Fig. 5, is it P-channel TFT? If yes, please use the absolute ID vs. VDS to express the current-voltage characteristics. Because the ON current is low, you can use the unit uA/um (ON current divided by channel width) to do the comparison with the drive current formed from the traditional TFTs. In Lines 36-37, the green laser as an annealing resource was useful to reduce the temperature, which the amorphous channel silicon can be transformed as poly-silicon and the channel mobility is increased hugely, such as this reference. You can consider it to richly increase your introduction part.
Ref.: Mu-Chun Wang, Hsin-Chia Yang, Hong-Wen Hsu, Zhen-Ying Hsieh, Shuang-Yuan Chen, Shih-Ying Chang, and Chuan-Hsi Liu, “Degradation Mechanism for Continuous-Wave Green Laser-crystallized Polycrystalline Silicon n-Channel Thin-Film Transistors under Low Vertical-Field Hot-Carrier Stress with Different Laser Annealing Powers,” Japanese Journal of Applied Physics (JJAP), vol. 50, pp.04DH16-1~4, Apr., 2011.
2. Could you provide the uniformity consequences? In fabrication, the uniformity is very important. They not only consider one device, but whole TFT panel. The more important device parameters are such as ID, Vth, and IG with error bar.
3. In Table 3, the Vth values show the negative and positive performance. You should explain them more, especially the physical mechanisms.
4. In Table 4, the gate leakage sensed at VGS=-5 V or -6V. What is the main reason? If the applied gate voltages are not the same, the gate leakage will show the large difference.
Author Response
ANSWER TO QUESTION 1:
I have changed the manuscript so that all of ID values are now |ID|/W values, as you requested. I have also changed all IG values to |IG| values.
I read the recommended JJAP paper, which was an excellent example of a TFT fabricated using polysilicon, innovatively using a CW green laser to anneal that polysilicon and improve its electrical properties. However, the JJAP’s fabrication process still uses high temperatures. As such, the JJAP paper does not contradict Lines 36-37 of our original manuscript, which states that “amorphous silicon and polysilicon are both incompatible with alternative low temperature solution based fabrication processes.” Regardless, a new sentence mentioning the JJAP paper as an example of TFT with a polysilicon channel has been added to our introduction, with the caveat that fabrication of that TFT still involves high temperature processes that are incompatible with low temperature solution based fabrication processes such those used in printed electronics technology (Lines 37-40). Moreover, the Introduction section has been expanded with further discussions regarding SWCNT based TFTs (Lines 54-83). The Discussion section has also been expanded by comparing our TFTs with those of Wang et al. (Lines 464-514).
ANSWER TO QUESTION 2:
This paper presents all of the TFTs we successfully fabricated and characterized without suffering a rapid dielectric breakdown. A larger dataset with more TFTs would indeed afford greater statistical significance. However, we unfortunately do not have the time to fabricate and characterize additional TFTs within the limited timeframe provided for revising the manuscript. We have also added additional material to the revised manuscript discussing the variation in SWCNT uniformity and density and how it affects devices performance (Lines 54-83). The variation in Vth is also now discussed (see our answer to your Question 3).
ANSWER TO QUESTION 3:
A new paragraph has been added to the Discussion section, to discuss the variability in Vth seen with our TFTs (Lines 492-515). This new paragraph discusses common reasons as to why Vth can vary considerably between SWCNT TFTs. This new paragraph also discusses a paper published by Wang et al. that discusses how the doping of SWCNTs can controllably finetune Vth in a fashion that is reliable and can reduce the Vth between TFTs:
Wang, H.; Wei, P; Li, Y; Han, J.; Lee, H. R.; Naab, B. D.; Liu, N.; Wang, C.; Adijanto, E.; Tee, B. C.-K.; Morishita, S.; Li, Q.; Gao, Y.; Cui, Y.; Bao, Z. tuning the threshold voltage of carbon nanotube transistors by n-type molecular doping for robust and flexible complementary circuits. PNAS 2014, 111(13): 4776-4781.
As this doping process is compatible with our own TFT fabrication process, it is suggested that we could incorporate such an additional SWCNT doping step into our fabrication process to finetune Vth and reduce the variation in Vth between TFTs.
ANSWER TO QUESTION 4:
The data collected for the four new TFTs of our manuscript used VGS = - 5 V. However, the two older TFTs from an earlier conference paper, included in Table 4 for comparison’s sake, used VGS = -6 V. However, you’re quite right that it’s necessary to consistently use the same VGS to make more direct comparisons. As such, I revisited the data for those two older TFTs are recomputed using VGS = -5 V. Consequentially, all of the values in Table 4 now use VGS = -5 V (Lines 432-434).
Round 2
Reviewer 1 Report
The manuscript has revised well according the reviewer;s comments. Theerefore, in my opinion, the article could be accepted to publish.
Reviewer 2 Report
I am glad to see that my concerns are addressed in the revised manuscript. It can be published in its current version.
Reviewer 3 Report
The revised article is great.